

# Effect of dietary concentrate to forage ratio on growth performance, rumen fermentation and bacterial diversity of Tibetan sheep under barn feeding on the Qinghai-Tibetan plateau

Hongjin Liu[1,2,3,*], Tianwei Xu[1,2,*], Shixiao Xu[1], Li Ma[1,2,3], Xueping Han[1,3,4], Xungang Wang[1,2,3], Xiaoling Zhang[1,2,3], Linyong Hu[1,2], Na Zhao[1,2], Yongwei Chen[4], Li Pi[1] and Xinquan Zhao[1,2]

[1] Northwest Institue of Plateau Biology, Chinese Academy of Science, Xining, China
[2] Key Laboratory of Adaptation and Evolution of Plateau Biota, Chinese Academy of Sciences, Xining, China
[3] University of Chinense Academy of Sciences, Beijing, China
[4] Technology Extension Service of Animal Husbandry of Qinghai, Xining, China
[*] These authors contributed equally to this work.

Corresponding author
Shixiao Xu, sxxu@nwipb.cas.cn

## ABSTRACT

This study aimed to research the effects of different dietary concentrate to forage (C:F) ratio on growth performance, rumen fermentation and bacteria diversity of barn feeding Tibetan sheep. The experiment contains fiver treatments (HS1, HS2 HS3, HS4 and HS5; $n = 8$, respectively) based on dietary C:F ratios 0:100, 15:85, 30:70, 45:55, and 60:40, respectively. The ruminal bacterial community structure was investigated through high-throughput sequencing of 16S rRNA genes in V4 hypervariable region. The results showed that increasing dietary concentrate feed level from 0% to 60% exerted a positive effect on DMI, BW gain, gain rate and feed conversation ratio ($F_{CR}$) in Tibetan sheep. The increases dietary concentrate feed level elevated$NH_3$-N, propionate and valerate concentrations, whereas, reduced molar ratio of acetate to propionate (A/P ratio) ($P < 0.05$). For rumen bacterial diversity, increases in dietary concentrate content contributed to lower alpha diversity indexes including Shannon wiener, Chao1 and observed species, meanwhile, significantly increased the abundances of the phylum *Bacteroidetes* and the genus *Prevotella_1* ($P < 0.05$). In conclusion, increases dietary concentrate content improved the growth performance and Tibetan sheep fed diets of 45% concentrate obtained a better performance; the inclusion of concentrate in feed changed rumen fermentation from acetate fermentation to propionate fermentation, and improved the energy utilization efficiency of Tibetan sheep; the increased in concentrate content significantly reduced rumen bacteria diversity and changed the abundance of some core bacteria.

## INTRODUCTION

Tibetan sheep (*Ovis aries*) live exclusively on the Qinghai Tibetan Plateau (QTP) with a altitude greater than 3,000 (*An, Dong & Dong, 2005*), they have adapted well to extremely harsh environment and ingested grasses as their sole source of nutrition (*Sasaki, 1994*; *Wiener et al., 2003*). Under traditional management, Tibetan sheep mainly grazed on natural pasture without concentrate supplementation, and always suffered seasonal live-weight variations and viciously cycled in "alive in summer, strong in autumn, thin in winter, tired in spring", due to seasonal fluctuations in natural pasture supply and the contradiction between herbage supply and livestock's requirement on the alpine pasture (*Dong et al., 2006*; *Xue, Zhao & Zhang, 2005*; *Xu et al., 2017*). During growing season (June to Oct.), natural pasture can provide enough herbage for livestock, the local people only drive livestock to alpine pasture for grazing. During non-growing season (Nov to May) and vegetation green-up periods, grazing livestock shared low performance mainly attributed to the decrement in herbage supply (both in quantity and quality). In addition, irrational grazing during this period destroyed regular growth of alpine plants and function recovery of alpine pasture. Previous studies have confirmed that spring short term rest-grazing is an effective measure for the functional recovery of alpine meadows (*Li et al., 2014*; *Li et al., 2017*; *Ma et al., 2017*), the local government encourages pastoralists to feed their livestock in warm shed during spring grazing break. However, few studies concerning growth performance of barn feeding Tibetan sheep during late non-growing and vegetation green-up periods were conducted, but which was quiet important for providing pastoralists guidance in livestock breeding.

The rumen microorganisms play an important role in the digestion of proteins, carbohydrates, starch, sugars and fats, which provides energy and proteins to the host by producing volatile fatty acids and bacterial proteins through anaerobic fermentation (*Ceconi et al., 2015*; *Jiang et al., 2015*) and finally affects ruminants performance. The rumen microorganism can be affected by many factors, such as diet, hosts and geographic region (*Clark, 1975*; *Lee et al., 2012*). The dietary nutrition level is a major factor affecting rumen microbial diversity, healthy status and production capacity of ruminants (*Clark, 1975*). The effect of diet on the structure of rumen microbial communities has been widely investigated in Yak, Tan sheep, Holstein dairy cows, Mehshana buffalo and goats (*Pitta et al., 2014*; *Jiao et al., 2015*; *Pitta et al., 2016*; *Wang et al., 2016*; *Xue et al., 2016*). However, as the dominant ruminant and living foundation for local herdsman, the studies of Tibetan sheep have only focused on the growth performance, slaughter performance and economic benefit under different dietary supplementation (*Baruah et al., 2012*; *Chen et al., 2015b*; *Dodd, Mackie & Cann, 2011*; *Feng et al., 2013*; *Lee et al., 2012*; *Xu et al., 2017*), but few comprehensive studies focused on the growth performance, rumen fermentation and rumen microbial communities of Tibetan sheep.

Therefore, this study aimed to investigate the growth performance, rumen fermentation and rumen bacterial diversity of Tibetan sheep under different dietary concentrate-to-forage ratios during late non-growing and green-up periods. We hypothesized that different C: F ratios could affect growth performance, rumen fermentation and rumen bacterial diversity

in Tibetan sheep. These results will be of great importance for providing guidance for local herdsmen in Tibetan sheep breeding and for future research on rumen microbial metabolism in Tibetan sheep.

## MATERIALS AND METHODS

### Animals and experimental design

This study was conducted at Haibei Demonstration Zone of Plateau Modern Ecological Animal Husandry Scientific and Technology in Haibei Prefecture, Qinghai Province, China (36°54′N, 100°56′E), from April 2016 to July 2016. All animal care procedures were consistent with the guidelines from the Institution of Animal Care and the Ethics Committee of the Northwest Institute of Plateau Biology, Chinese Academy of Sciences (NWIPB20160302). The processing of the samples after collection was performed strictly in accordance with the guidelines of NWIPB.

A total of 40 female yearling Tibetan sheep with familiar body conditions (21.39 ± 1.18 kg BW, and not under current antibiotic treatment) were randomly divided into five treatments (8 in each) under different C:F ratios (on dry matter basis):HS1(0:100), HS2 (15:85), HS3 (30:70), HS4 (45:55) and HS5 (60:40), respectively. All sheep were allocated into 5 pens within a warm shed. The concentrate and oat hay were manually mixed and fed (dry matter) based on 3.5% BW of Tibetan sheep. Before study, all the animals were fed the experimental diet for a 15-day adaptation phase. During formal 80 day experiment, the experimental animals were separately fed twice dairy, at 8:00 am and 5:00 pm. The diets, salt brick and clean water were available throughout the entire experiment. The ingredients and nutrient compositions of the animal diet are shown in Table 1 and Table S1. The concentrate feed used in this study was produced by Menyuan Yongxing Ecological Agricultural Development Co., Ltd., and oat hay are harvested, bundled and stored in the year 2015.

### Measurement and sampling

The diets and oats in each treatment were weighed and recorded daily to estimate individual dry matter intake (DMI). Diet samples of approximately 500 g from each treatment were collected, dried at 60 °C, ground through a 1-mm sieve and stored in a vacuum dryer for nutritional analysis. To measure animal growth performance, Tibetan sheep were weighed before feeding using an electronic balance at the beginning and end of formal experiment. We selected four Tibetan sheep from each treatment group for rumen fluid sampling. Specifically, on 80th day, rumen content samples were obtained 3–4 h after the morning feeding using a stomach tube attached to an electric pump, which has been confirmed to yield similar results for rumen microorganisms and fermentation parameters as sampling using a rumen cannula (*Ramos-Morales et al., 2014*). The rumen contents were filtered through four layers of sterilized gauze. Approximately 67 mL of liquid was obtained from the rumen of each Tibetan sheep. The rumen fluid was separated into three samples, one (approximately 2 mL) was transferred into sterilized freezing tubes and stored in liquid nitrogen for DNA extraction; the second sample (approximately 15 mL) was immediately used to measure the pH; and the third sample (approximately 50 mL) was used for the
**Table 1** The nutritional composition of the whole diets among different treatments (% DM basis).

| Items | Groups[a] | | | | |
|---|---|---|---|---|---|
| | HS1 | HS2 | HS3 | HS4 | HS5 |
| Ingredient (%) | | | | | |
| Concentrate feeds[c] | 0 | 15 | 30 | 45 | 60 |
| Oats hay | 100 | 85 | 70 | 55 | 40 |
| Nutrient content | | | | | |
| DM[b] | 93.42 | 92.70 | 91.82 | 91.23 | 90.35 |
| CP | 6.31 | 7.69 | 9.37 | 10.38 | 11.94 |
| Starch (mg/g) | 25.58 | 26.95 | 32.55 | 33.72 | 39.37 |
| EE | 2.13 | 2.21 | 2.36 | 2.37 | 2.49 |
| ADF | 34.1 | 30.68 | 27.33 | 23.79 | 19.64 |
| NDF | 57.64 | 52.04 | 44.85 | 39.05 | 33.12 |
| Calcium | 0.35 | 0.40 | 0.48 | 0.55 | 0.61 |
| Magnesium | 0.24 | 0.27 | 0.32 | 0.35 | 0.42 |
| Phosphorus | 0.22 | 0.27 | 0.33 | 0.38 | 0.45 |

**Notes.**

[a]The tratments HS1, HS2, HS3, HS4 and HS5 refer to the C:F ratios of 0:100, 15:85, 30:70, 45:55 and 60:40, respectively.

[b]DM, dry matter; CP, crude protein; EE, ether extract; ADF, acid detergent fibre; NDF, neutral detergent fibre.

[c]Manufactured by Menyuan Yongxing Ecological Agricultural Development Co., Ltd. Contained corn (48%), wheat (30%), soybean meal (7%), colza cake (6%) cottonseed meal (5%), salt (1%), pre-mix (1%), $CaHPO_4$ (1%) and $CaCO_3$ (1%).

assessment of rumen fermentation parameters, including ammonia nitrogen ($NH_3$-N) and rumen volatile fatty acids (VFAs).

## Chemical analysis and calculation

The feed samples were fried in an oven at 135 °C for 3 h to obtain the DM (AOAC, 1990; Method No. 930.15). The total N was detected using Kjeldahl method; the crude protein content was calculated as $6.25 \times N$ (Method No. 984.13); the ether extract (EE) was measured using the Soxhlet system (Method No. 954.02); the acid detergent fiber (ADF) and neutral detergent fiber (NDF) of diet were analysed using method described by *Soest, Robertson & Lewis (1991)*; and the starch was measured according to PRC national standard (GB 5009.9-2016).

The body weight gain (BW gain), gain rate, average daily BW gain (ADG) and feed conversion ratio ($F_{CR}$) were calculated according to the following equations:

$$BW_{gain}(kg) = BW_f - BW_i$$

$$Gain\ rate\ (\%) = 100 \times (BW_f - BW_i)/BW_i$$

$$ADG\ (g/d) = 1000 \times (BW_f - BW_i)/t$$

$$F_{CR} = DM_{consume}/(BW_f - BW_i)$$

where $BW_f$ and $BW_i$ are the final and initial body weights (kg), respectively, t is the experimental time (d), and $DM_{consume}$ is the total feed consumed during the experiment (kg DM). The ruminal fluid pH was measured using a portable pH meter (PHSJ-3F; Precision Instruments Company, Shang Hai, China). For VFAs measurements, the rumen

fluid was centrifuged at 12,000 g for 15 min, and the VFAs in the supernatant were analysed using a gas chromatograph Agilent 7890A (Agilent Technologies Inc., Santa Clara, CA, USA) equipped with a polar capillary column (DB-WAX, 30 m × 0.25 mm × 0.25 μm) and a flame ionization detector (FID, temperature set at 250 °C). The temperature-programmed conditions were as follows: the temperature was maintained at 60 °C for 2 min, increased to 180 °C at rate of 10 °C/min, and increased to 250 °C at rate of 20 °C/min; the shunt ratio was 20:1; the flow rate was 1 mL/min; and the inlet temperature was 200 °C. The $NH_3$-N content in the supernatant was quantified using a continuous flow analyser (SEAL Auto Analyser 3, SEAL Analystical, Norderstedt, Germany) described by *Rhine et al. (1998)*.

## DNA extraction and PCR amplification

Genomic DNA was directly extracted from 0.2 g of each semisolid-state sample using cetyl trimethylammonium bromide (CTAB) method (*Porebski, Bailey & Baum, 1997*). The DNA quality was assessed via 2% agarose gel elcectrophoresis, and metagenomic DNA concentratios were determined with a NanoDrop 2000 (Thermo Fisher, Waltham, MA, USA). The DNA was then diluted to 1 ng/μL using sterile water and stored at −4 °C for PCR amplification.

For PCR amplification, the V4 hypervariable region of the bacterial 16S rRNA gene was amplified using the universal primers 515F (5′-GTGCCAGCMGCCGCGGTAA-3′) and 806R (5′- GGACTACHVGGGTWTCTAAT-3′) with unique barcodes. PCRs were performed in 25 μL reactions consisting of 12.5 μL Phusion® High-Fidelity PCR Master Mix (New England Biolabs, Ipswich, MA, USA), one μL of forward and reverse primers, one μL of template DNA and 9.5 μL of autoclaved distilled water. The thermal cycling program consisted of initial denaturation at 94 °C for 3 min, 30 cycles of denaturation at 94 °C for 40 s, annealing at 56 °C for 1 min, and elongation at 72 °C for 1 min and a final incubation step at 72 °C for 10 min. For PCR product quantification and qualification, we obtained mixtures of the same volumes of 1× loading buffer (containing SYBR Green from Shanghai, SanoBio) and the PCR products were exmined ona 2% agarose gel electrophoresis. Samples with a bright band between 400 and 450 bp were mixed and purified using a Qiagen Gel Extraction Kit (Qiagen, Germany). Sequencing libraries were then generated using a TruSeq® Ultra^TM DNA Library Prep Kit for Illumina (NEB, Ipswich MA, USA) following the manufacturer's recommendations, and the index codes were then added. The library quality was assessed using a Qubit® 2.0 fluorometer (Thermo Scientific) and an Agilent Bio analyzer system. The library was sequenced on an Illumina HiSeq PE250 platform (Novogene, Beijing, China).

## Analysis of sequencing data

The paired-end reads were assigned to samples based on their unique barcodes, and the barcodes and primer sequences were then trimmed. The raw reads were filtered according to the following rules: Removing reads containing more than 10% of unknown nucleotides (N); removing reads containing less than 80% of bases with quality ($Q$-value) >20. The FLASH (version 1.2.7) was then used to merge paired-ends reads as raw tags with a minimum overlap of 10bp and mismatch error rates of 2% (*Magoc & Salzberg, 2011*). The
noisy sequences of raw tags were filtered by QIIME (version 1.9.1) pipeline to obtain the high-quality clean tags (*Caporaso et al., 2010*). The reads were then compared with a Gold database using the UCHIME algorithm to detect and remove the chimaera sequences (*Edgar et al., 2011*). All the sequences were analysed using Uparse software (version 7.0.1001), and sequences with greater than or equal 97% similarity were assigned to the same operational taxonomic unit (OTU). Taxonomic information for each representative sequence was annotated using the Greengenes database based on the RDP classifier algorithm (version 2.2) (*Wang et al., 2007*).

## Statistical analysis

Alpha and species diversity indexes, including the Shannon-Wiener indexe, Chao1 index, Good's coverage and observed species were calculated by Qiime and graphed using Origin (version 8.0). The correlation among growth performance, rumen fermentation parameters and bacteria diversity were analyzed using SPSS (version 17.0), and heat map of correlations were also prepared using Origin. The beta diversity based on weighted UniFrac distance, was visualized through a principal coordinate analysis (PCoA). One-way ANOVA with Tuckey's test was performed to compare the differences in relative abundances among different treatments, and Duncan's multiple comparison test was used to determine the effects of dietary C: F ratios on BW gain, gain rate, ADG, DMI, DM consume and $F_{CR}$ using SPSS (version 17.0). Effects were considered significant at $P < 0.05$. The results are shown as the means $\pm$ SEMs.

# RESULTS

## Growth performance of Tibetan sheep

As shown in Table 2, treatments HS4 and HS5 significantly increased BW gain, gain rate and ADG, whereas reduced $F_{CR}$ as compared to treatments HS1 and HS2 ($P < 0.05$). Tibetan sheep fed diet in group HS1 showed the lowest growth performance. No significant difference were detected in BW gain and $F_{CR}$ between treatments HS4 and HS5, and the initial BW showed no notably difference among the five groups ($P = 0.196$).

## Rumen fermentation parameters

As shown in Table 3, the $NH_3$-N, propionate and valerate concentrations in treatments HS4 and HS5 were significantly higher than that of other three groups ($P < 0.05$). There was a decreasing trend for total VFA, acetate and isovalerate concentrations, but no significant differences were detected among the groups. The increases in dietary concentrate level significantly decreased A:P ratio ($P < 0.05$). The ruminal pH tended to decrease, but no significant difference was detected ($P > 0.05$), and the average pH value was approximately 6.61.

## Sequencing and taxonomic composition of the rumen bacterial community

A total of 1,497,607 PE reads were obtained, and 1,461,673 clean reads from 20 different samples were generated after quality control (Table S2). All the sequences were aligned and clustered into OTUs using a 97% sequence identity as a cut-off, which yielded a

**Table 2  The effects of different dietary C: F ratios on the growth performance of Tibetan sheep.**

| Items | Groups[a] | | | | | p-value |
|---|---|---|---|---|---|---|
| | HS1 | HS2 | HS3 | HS4 | HS5 | |
| Initial BW (kg) | 21.31 ± 0.92 | 22.13 ± 3.06 | 21.25 ± 1.69 | 19.88 ± 1.1.81 | 22.19 ± 2.34 | 0.196 |
| Final BW(kg) | 23.81 ± 1.25[d] | 26.44 ± 3.07[cd] | 28.38 ± 2.23[bc] | 29.75 ± 2.62[b] | 32.94 ± 3.49[a] | <0.01 |
| BW gain (kg) | 2.50 ± 0.76[d] | 4.31 ± 1.92[c] | 7.13 ± 1.71[b] | 9.88 ± 1.43[a] | 10.75 ± 1.79[a] | <0.01 |
| Gain rate (%) | 11.73 ± 3.50[d] | 19.93 ± 6.54[c] | 33.78 ± 8.63[b] | 49.98 ± 7.13a | 48.59 ± 7.56[a] | <0.01 |
| ADG (g/d) | 31.25 ± 9.45[d] | 53.91 ± 12.91[c] | 89.06 ± 21.33[b] | 123.44 ± 17.91[a] | 134.38 ± 22.41[a] | <0.01 |
| DMI (g/d | 670.30 ± 45.26[c] | 719.16 ± 120.09[bc] | 741.21 ± 71.22[abc] | 807.68 ± 32.29[ab] | 851.93 ± 98.86[a] | <0.01 |
| DM consume (kg) | 53.62 ± 3.62[c] | 57.53 ± 8.98[bc] | 59.30 ± 5.70[bc] | 64.61 ± 4.98[ab] | 68.15 ± 7.91[a] | <0.01 |
| FCR | 23.86 ± 9.75[a] | 13.94 ± 3.38[b] | 8.92 ± 3.06[c] | 6.67 ± 1.15[c] | 6.44 ± 1.00[c] | <0.01 |

Notes.
[a]The treatments HS1, HS2, HS3, HS4 and HS5 refer to concentrate-to-forage ratios of 0:100,15:85, 30:70, 45:55 and 60:40. DMI, dry matter intake; DM consume, DMI ± t, t is experimental time (d), BW, body weight; ADG, average daily BW gain. FCR, feed conversion ratio. Values in the same row with different superscripts are significantly different (P < 0.05).

**Table 3  The effects of different dietary C: F ratios on the rumen fermentation parameters of Tibetan sheep.**

| Parameters | Groups[a] | | | | | P[b] |
|---|---|---|---|---|---|---|
| | HS1 | HS2 | HS3 | HS4 | HS5 | |
| Ammonia nitrogen (mg/L) | 46.20 ± 12.37[c] | 46.20 ± 12.37[c] | 64.00 ± 3.88[b] | 67.10 ± 8.21[b] | 107.06 ± 15.07[a] | <0.05 |
| Acetate (%) | 77.82[a] | 71.70[b] | 64.10[b] | 60.14 ± 8.55[b] | 64.55 ± 2.15[b] | 0.32 |
| Propionate (%) | 12.60 ± 0.71[b] | 16.02 ± 0.18[b] | 22.79 ± 1.13[a] | 26.81 ± 3.60[a] | 29.38 ± 0.95[a] | <0.05 |
| Butyrate (%) | 8.25 ± 0.38 | 10.66 ± 1.73 | 11.30 ± 0.40 | 11.51 ± 2.20 | 12.02 ± 0.40 | 0.23 |
| Isovalerate (%) | 0.79 ± 0.10[a] | 0.92 ± 0.07[a] | 0.83 ± 0.02[a] | 0.58 ± 0.03[b] | 0.72 ± 0.09[b] | 0.23 |
| Valerate (%) | 0.52 ± 0.02[b] | 0.70 ± 0.04[b] | 0.98 ± 0.03[b] | 1.11 ± 0.04[a] | 1.50 ± 0.12[a] | <0.05 |
| Total volatile fatty acids (mmol/L) | 74.94 ± 10.76 | 66.12 ± 2.95 | 65.21 ± 1.00 | 67.44 ± 1.56 | 61.24 ± 7.67 | 0.11 |
| Acetate to propionate ratio (A:P) | 6.38 ± 1.55[a] | 4.47 ± 0.16[a] | 2.85 ± 0.24[b] | 2.24 ± 0.39[b] | 2.19 ± 0.69[b] | <0.05 |
| pH | 6.86 ± 0.01 | 6.83 ± 0.03 | 6.83 ± 0.04 | 6.63 ± 0.06 | 6.42 ± 0.10 | 0.41 |

Notes.
[a]The treatments HS1, HS2, HS3, HS4 and HS5 refer to concentrate to forage ratios of 0:100, 15:85, 30:70, 45:55 and 60:40, respectively.
[b]Values in the same row with different superscripts are significantly different (P < 0.05).

total of 49,216 OTUs (Table S2). Rarefaction curves were generated with a 3% cut of for comparisons of species and richness as shown in Fig. 1. All the curves of the observed species numbers tended to saturate to a plateau at the minimum sequence number of 60,000 tags. In addition, the HS4 and HS5 groups tended to cluster together with a lower observed species number than the HS1, HS2 and HS3 groups.

The taxonomic analysis detected a total of 41 phyla (Table S3), and the most abundant phyla were *Bacteroidetes* (52.18%), *Proteobacteria* (20.34%) and *Firmicutes* (14.34%). The top ten phyla, which exhibited the highest relative abundances, were prevalent in all the samples, accounting for nearly 98% of all sequences (Fig. 2A). The moderate abundant phyla included *Fibrobacteres* (1.25%), *Cyanobacteria (1.04%)*, *Acidobacteria* (0.96%),*Tenericutes* (1.49%), *Actinobacteria* (0.96%) and *Verrucomicrobia* (0.47%). The other known phyla accounted for 1.48%, whereas the remaining sequences that were unclassified accounted for 0.26%.

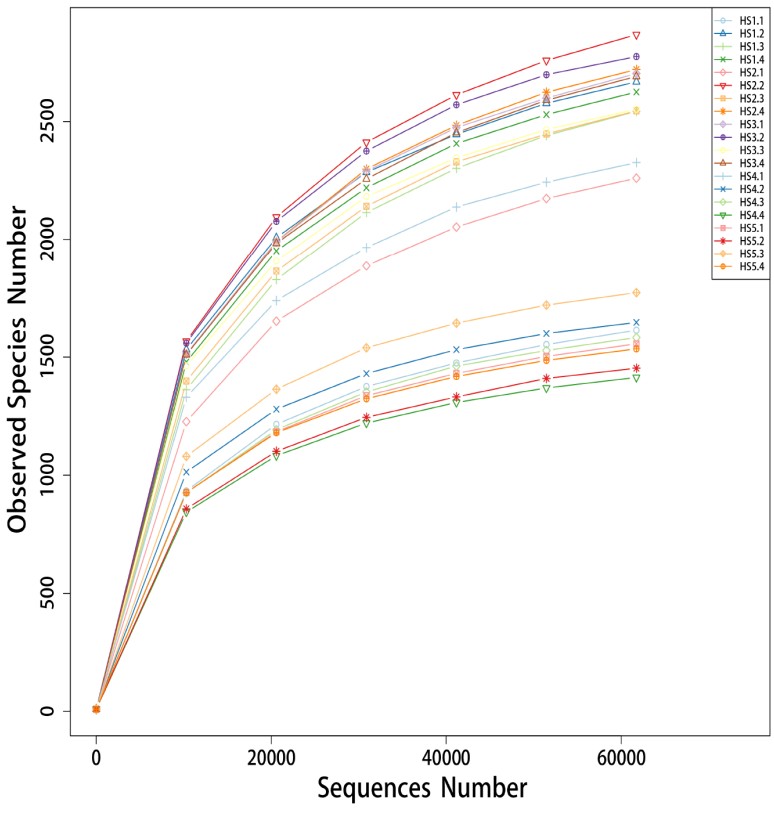

**Figure 1** **The rarefaction analysis anomg the 20 different samples.** The group HS1 samples included sheep HS1.1, HS1.2, HS1.3 and HS1.4. The treatment HS2 samples included sheep HS2.1, HS 2.2, HS2.3 and HS4. The treatment HS3 samples included sheep HS3.1, HS3.2 HS3.3 and HS3.4. The treatment HS4 samples included HS4.1, HS4.2, HS4.3 and H4.4. The treatment HS5 samples included HS5.1, HS5.2, HS5.3 and H5.4. The treatment HS1, concentrate to forage ratio 0:100; HS2, concentrate to forage ratio 15:85; HS3, concentrate to forage ratio 30:70; HS4, concentrate to forage ratio 45:55; HS5, concentrate to forage ratio 60:40.

At the genus level, we detected 129 distinct genera across all the groups (Table S4), and 10 genera whose relative abundances more than 1% were perceived as the most important bacteria that affecting the rumen environment and digestive system (Fig. 2B). The genus with relatively high across all the samples were *Prevotella_1* (26.81%), *Succinivibrionaceae_UCG-002* (7.11%), *Ruminobacter* (6.03%), *Rikenellaceae_RC9_gut_group* (4.73%), *Prevotellaceae_UCG-003* (2.65%). Minor genera, such as *Prevotellaceae_UCG-001*, *Erysipelotrichaceae_UCG-004*, *Fibrobacter*, *Christensenellaceae_R-7_group* and *Ruminococcaceae_NK4A214_group* accounted for 1.23%, 1.09%, 1.01%, 1.22 and 1.09% of the sequences, respectively. The other known genera accounted for 11.32% of the sequences, whereas sequences that were unclassified accounted for 34.23% of the sequences.

### Effect of diets with different C:F ratios on the bacterial community

To determine alpha diversity, we calculated Shannon-Wiener, Chao 1, observed species and Good's coverage indexes, as shown in Fig. 3. The indexes of Shannon-Wiener, Chao

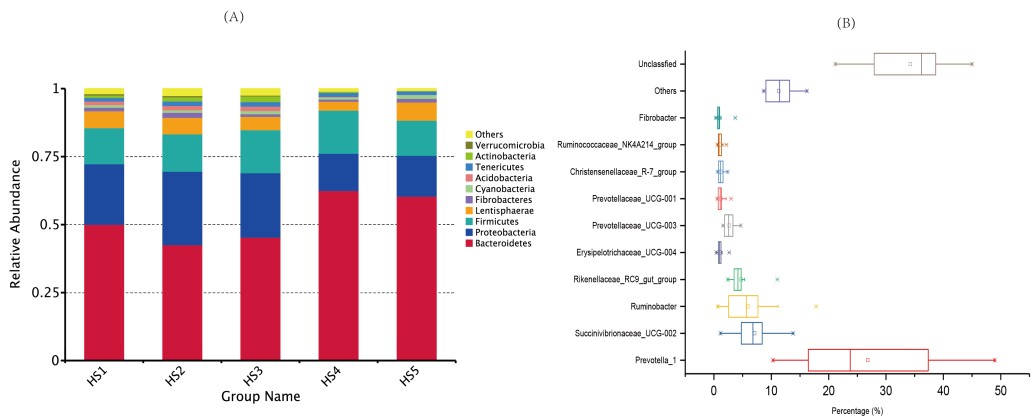

**Figure 2  Dominant bacterial phylum in individual samples and the shared genera across the ruminal samples.** (A) The top 10 shared bacterial taxonomic composition across ruminal samples at the phylum level. (B) The top 10 shared bacterial toxonomic composition across ruminal samples. Percentage is shown on the X-axis. The box represent the interquartile range between the first and the third quartiles, and the symbol "— —" represent the max value, "×" represent the variation range from 1% to 99% and "□" represent the mean value. The treatment HS1, concentrate to forage ratio 0:100; HS2, concentrate to forage ratio 15:85; HS3, concentrate to forage ratio 30:70; HS4, concentrate to forage ratio 45:55; HS5, concentrate to forage ratio 60:40.

1 and observed species in treatments HS1, HS2 and HS3 were significantly higher than treatments HS4 and HS5 (Figs. 3A, 3B and 3C). Good's coverage among five treatment were greater than 99% (Fig. 3D). As shown in Fig. 4, the PCoA result showed that the rumen bacterial communities of the five treatments were mainly classified into three clusters (Fig. 4). The HS1 and HS2 treatments clustered very closely together, the HS4 and HS5, and the HS3 treatments formed two clusters. In addition, HS3 treatment were closer to the cluster composed of the HS1 and HS2 treatments, which only represented 7.14% of the variability obtained with PC2.

The relative abundance of bacterial taxa was used to describe the impact of diets with different C:F ratios on bacterial community. At the phylum level, the top eight phylum (relative abundance >1%) were analyzed in Table S5. The ruminal compositions of the HS1 to HS3 treatments contained a significantly lower relative abundance of *Bacteroidetes*, whereas there were higher relative abundances of *Proteobacteria*, *Acidobacteria*, and *Actinobacteria* than those of the HS4 and HS5 treatments ($P < 0.05$). No significant differences were found in the other phyla ($P > 0.05$). At the genus level, the top 12 genera (relative abundance >1%) were analyzed in Table S6. Treatments HS4 and HS5 shared a higher relative abundance of *Prevotella_1* than the other three treatments ($P < 0.05$). No significant differences were found in the rest of genera ($p > 0.05$).

## Relationships between rumen bacterial diversity and host growth performance

As shown in Fig. 5, a significant negative correlation was found between bacterial diversity (including the Shannon-Wiener, Chao1, and observed species indexes) and growth performance (including BW gain, ADG, and DMI) with increases in the C:F ratio

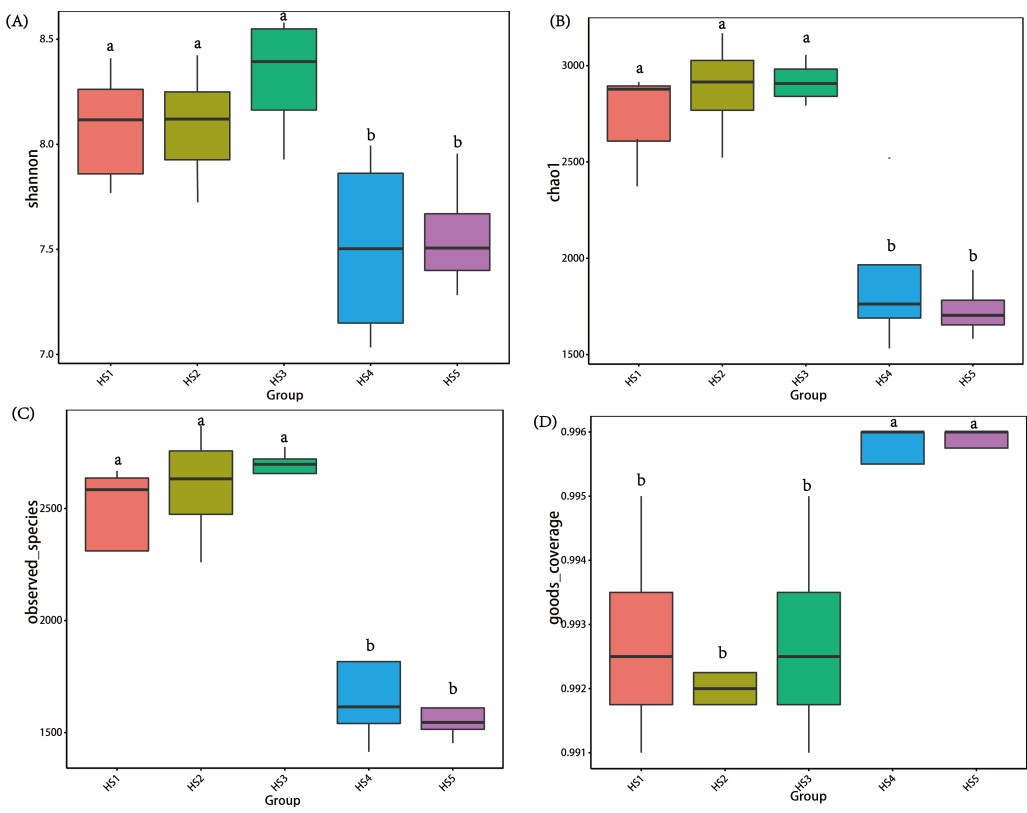

**Figure 3** **The differences in bacteria community diversity and richness indexes among different feed-ing treatment.** (A) The Shannon-Wiener diversity among each treatment. (B) The Chao1 richness estima-tor among each treatment. (C) The observed species in each treatment. (D) Good's coverage in samples among each treatment. The treatment HS1, concentrate to forage ratio 0:100; HS2, concentrate to forage ratio 15:85; HS3, concentrate to forage ratio 30:70; HS4, concentrate to forage ratio 45:55; HS5, concen-trate to forage ratio 60:40.

($P < 0.05$). $F_{CR}$ showed a negative correlation with increases in the DMI, BW gain and ADG and a positive correlation with the Shannon-Wiener, Chao1 and observed species indexes. In addition, growth performance showed significant positive correlations with increases in the dietary C:F ratio ($P < 0.05$).

## Relationships between rumen fermentation parameter and microorganisms

A heat map of the correlations between the top 45 genera (relative abundance >0.1%) and rumen fermentation was constructed (Fig. 6). A total of 360 correlation coefficients were generated, and 33 of these coefficients (9.17% of the total correlation coefficients) showed significant correlations ($P < 0.05$). The fermentation parameters pH, acetate, isovalerate and A:P were significantly positively correlated with most genera (e.g., *Victivalls*, *Thalassospira* and *Sphaerochaeta*), whereas valerate and $NH_3$ showed a negative correlation with most genera (e.g., Victivalls, *Ruminococcaceae_UCG_02*, and *Thalassosphira*). *Prevotella_1*, which was the most abundant genera, was significantly positively ($P < 0.05$)

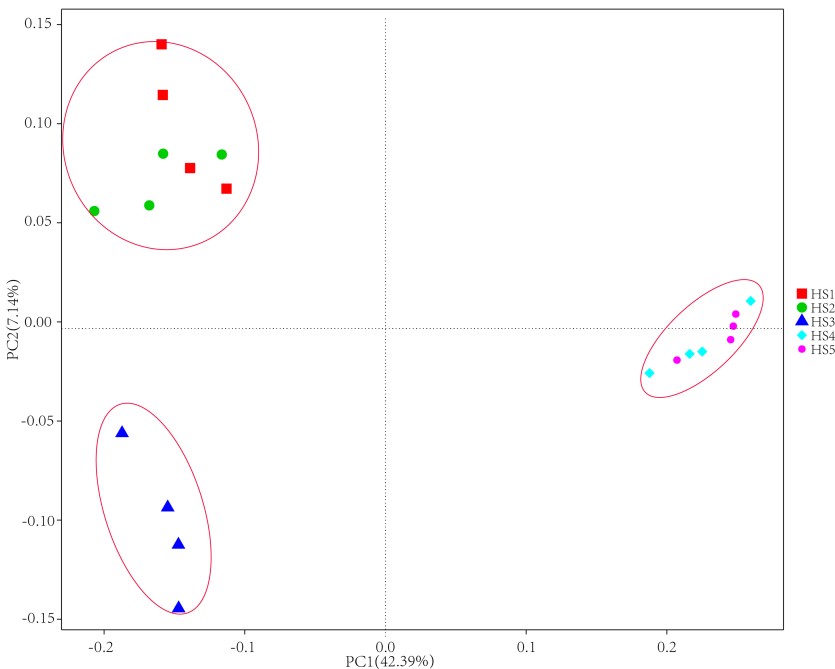

**Figure 4  The principal coordinate analysis (PCoA) using weighted UniFrac metrics.** The treatment HS1, concentrate to forage ratio 0:100; HS2, concentrate to forage ratio 15:85; HS3, concentrate to forage ratio 30:70; HS4, concentrate to forage ratio 45:55; HS5, concentrate to forage ratio 60:40.

correlated with $NH_3$, propionate and valerate and negatively correlated with pH. No significant correlation were detected between butyrate and the top 45 genera.

## DISCUSSION

### Growth performance of barn-fed Tibetan sheep

Previous studies mainly aimed to ensure that Tibetan sheep live through grazing breaks (*Ma, 2008*; *Xie et al., 2014*), and there is little knowledge on the BW gain and feed efficiency of Tibetan sheep fed diets with different C:F ratios during late non-growing and vegetation green-up periods. Our results revealed that increases in dietary concentrate feed levels within a certain range (0% to 45%) exerted a positive effect on the feed intake, BW gain and $F_{CR}$ of Tibetan sheep during spring grazing breaks. This finding can be mainly attributed to the fact that high-concentrate diet contains more digestive energy and nonstructural carbohydrates (*Wang et al., 2015*), which leads to increased nutrient intake, faster digestion through the digestive tract, and then improved growth performance and higher feed efficiency (*Haddad & Ata, 2009*; *Haddad, 2005*).

Under traditional grazing, Tibetan sheep exhibit low growth performance (ADG = 36∼55 g/d) during spring, and more seriously, unsustainable grazing during this period decreases the plant species richness and biomass in alpine pasture (*Ma, 2008*). In contrast, shortened rest-grazing increases the standing herbage biomass by 77%∼189% (*Li et al., 2017*; *Ma et al., 2017*). Herein, we confirmed that a diet with diet with a 45% concentration

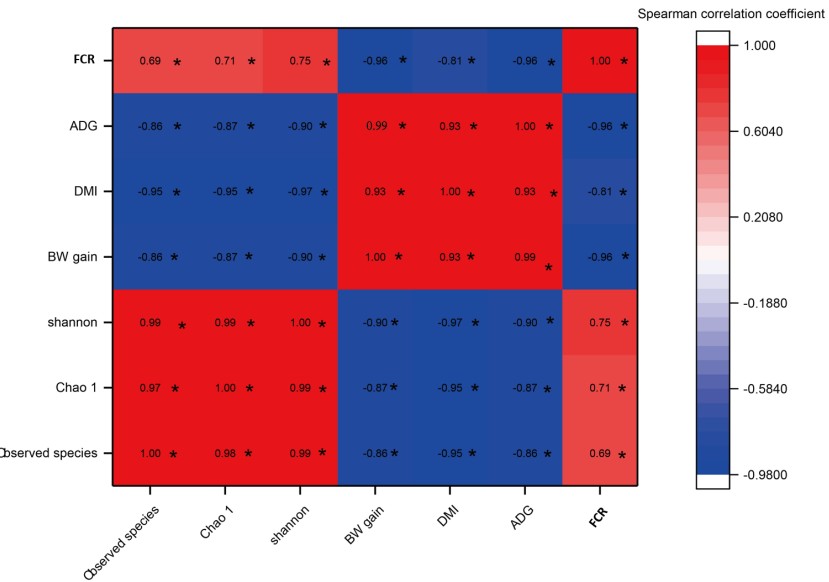

**Figure 5** **The heat map of the correlations between bacterial diversity and growth performance of Tibetan sheep.** DMI, dry matter intake; ADG, average daily body weight gain; FCR, feed conversion ratio; BW again, body weight gain. The treatment HS1, concentrate to forage ratio 0:100; HS2, concentrate to forage ratio 15:85; HS3, concentrate to forage ratio 30:70; HS4, concentrate to forage ratio 45:55; HS5, concentrate to forage ratio 60:40. *Significant at $P < 0.05$.

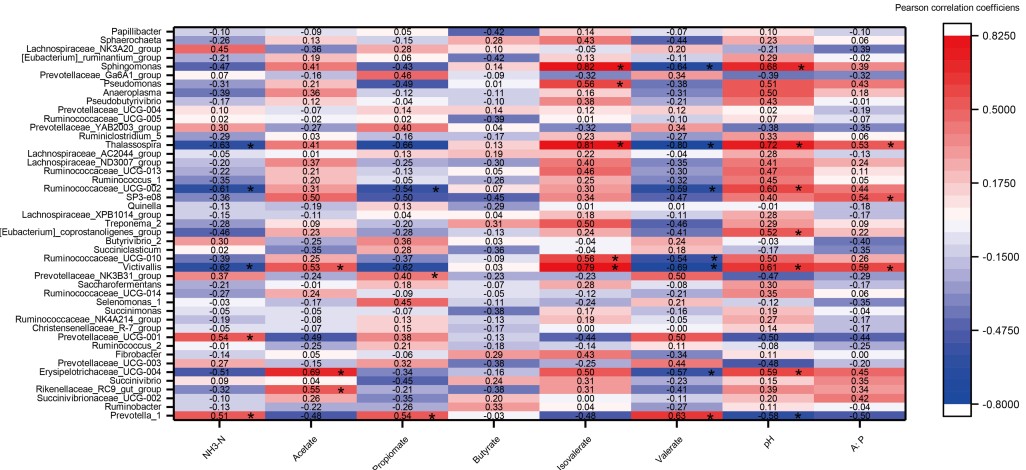

**Figure 6** **The heat map of the correlations between bacterial diversity and growth performance of Tibetan sheep.** DMI, dry matter intake; ADG, average daily body weight gain; FCR, feed conversion ratio; BW again, body weight gain. The treatment HS1, concentrate to forage ratio 0:100; HS2, concentrate to forage ratio 15:85; HS3, concentrate to forage ratio 30:70; HS4, concentrate to forage ratio 45:55; HS5, concentrate to forage ratio 60:40. *Significant at $P < 0.05$.

supplementation significantly improves the performance of Tibetan sheep. Therefore, the use of spring rest-grazing combined with barn feeding could increase livestock performance and promote the functional recovery of alpine pasture.

## Effect of different C: F ratios on rumen fermentation parameters

The pH, $NH_3$-N concentration and VFA molar ratio are the main internal environmental indicators of rumen fermentation. If the rumen pH was in the range of 6.2 to 7.0, the ecological environment of rumen microorganisms could be relatively stable, which could ensure normal rumen fermentation (*Weng, 2013*; *Zhang et al., 2013*). In our study, the ruminal pH ranged from 6.42 to 6.86, and this finding indicated that the increased proportion of concentrate did not induce acidosis, which is usually defined as a decrease in the rumen pH to less than the threshold value of 6.0 (*Nocek, 1997*). The ruminant pH values obtained in the present study are inconsistent with previous studies (*Cerrillo, Russell & Crump, 1999*; *Chen et al., 2015a*). The pH value can be affected by many factors, such as higher doses of flour or starches (*Lettat et al., 2010*; *Minuti et al., 2014*), the time from concentrate feed supplementation to ruminal liquid collection (*Lettat et al., 2010*), and the insertion depth of the rumen catheter (*Li et al., 2009*). In our study, increases in the dietary concentrate level tended to decrease the ruminal pH, but no significant difference was detected ($P > 0.05$), which could be due to the starch content. The HS1 and HS2 groups had a lower starch content (25.58 and 26.95 mg/g, respectively), which led to a higher pH value, whereas the HS3, HS4 and HS5 groups had a higher starch content (32.55, 33.72 and 39.37mg/g, respectively), which resulted in a relatively lower pH value. The dietary C:F ratios did not significantly affect the total volatile fatty acids, which is in accordance with previous research (*Chen et al., 2015a*). This result might be due to the ability of the rumen system that can adapt to appropriate dietary C:F ratios through the self-adjustment of rumen microorganisms. Ammonium nitrogen which is the final product through the decomposition of protein and nonprotein in feed, and it is also the main nitrogen source for the synthesis of bacterial protein by rumen microorganisms. In our study, high concentrate feed level elevated NH3-N concentration, which is consistent with the former study (*Reddy & Reddy, 1985*; *Yang, Beauchemin & Rode, 2001*). The high dietary concentrate increased nitrogen content in the rumen, thereby increasing ruminal ammonia nitrogen (*Moorby et al., 2006*). As for individual VFA concentrations in current study, diets with high-concentrate levels slightly reduced acetate concentration, whereas significantly increased propionate proportion, thereby resulting in a significant reduction of A:P ratio. (*Andrade & Schmidely, 2006*; *Polyorach, Wanapat & Cherdthong, 2014*). From the point of view of energy utilization, a decrease in the A:P ratio reflects an improvement in the feed energy utilization efficiency, which also explains the improved growth performance of Tibetan sheep in the high-concentrate groups (HS4 and HS5).

## Core bacterial communities in the rumen

Although rumen microbial community composition varies with diet and host, a 'core microbiome' is found across a wide geographical range (*Henderson et al., 2015*). The dominance of *Bacteroidetes* or *Proteobacteria* could be attributable to variations in diet,

environment, hosts and farming practices over a wide geographical range (*Amato et al., 2013*; *Henderson et al., 2013*). In our study, different dietary concentrate level induced *Bacteroidetes* (55.02%) to become the most abundant phyla, followed by *Proteobacteria* (22.10%), and this microbial distribution of major phyla was similar to that obtained in previous studies on Yaks (*Chen et al., 2015a*). These results might be related to the functions of rumen bacteria. Members of the phylum *Bacteroidetes* have a greater ability to degrade protein and carbohydrates than species belonging to *Proteobacteria* (*Huo, Zhu & Mao, 2014*; *Pitta et al., 2014*). Our study also found that the relative abundance of *Firmicutes* was the third largest phylum and was lower than that of *Proteobacteria*. Similar results were observed in bovines and cattle during the transition from forage to concentrate (*Jami et al., 2013*). However, *Xue et al. (2016)* observed that the most abundant phyla in the rumen of natural pasture-grazing Tibetan sheep are *Bacteroidetes*, followed by *Firmicutes*. One possible reason for this difference is that the hosts investigated by *Xue et al. (2016)* were grazed on natural pasture and that the forage nutrition and types differed from those in our study, and these differences might have resulted in a higher abundance of *Firmicutes*. Additionally, the phylum *Fibrobacteres*, with an abundance less than 1.5%, comprised only a small fraction of the community composition compared with those of other phyla, and these results agree with previous studies (*Petri et al., 2013*; *Wang et al., 2016*).

In our study, although different C:F ratio did not change the core structure of the rumen microbiome, the relative abundances of *Bacteroidetes* and *Proteobacteria* showed noticeable shifts at phylum level. Tibetan sheep fed high-concentrate diets significantly increased the relative abundance of *Bcateroidetes*, whereas, reduced the relative abundance of *Proteobacteria*. Previous research revealed that *Bacteroidetes* were the major rumen microorganisms in degrading non-fibrous carbohydrates and contained genes related to the degradation of non-fibrous polysaccharides (*Russell & Diez-Gonzalez, 1997*). In our study, high concentrate feed contained more non-fibrous carbohydrates and polysaccharides, thus increasing the abundance of *Bacteroidetes*. At the genus level, *Prevotella_1* was the predominant genus, and the relative abundances of this genus significantly increased with increases in dietary C:F ratio. This finding was consistent with that previously obtained by *Stevenson & Weimer (2007)*. The high abundance of this genus can perhaps be explained from two points of view. Firstly, it if possible that this bacterial genus has a wide metabolic niche due to genetic relatedness or to the high genetic variability that enable this genus to occupy different ecological niches within the rumen. Secondly, *Prevotella_1* strains play an important role in the degradation and utilization of plant noncellulosic polysaccharides, protein, starch and xylans. The increase in the abundance of this genus could be attributed to dietary nutrition changes, such as increased dietary protein and starch. In addition, *Prevotella_1* is also considered to be associated with propionic acid production (*Strobel, 1992*), which might also explain the increasing proportion of propionate obtained with increase in the C:F ratio.

### Relationship between bacterial diversity and growth performance of Tibetan sheep

In the past 50 years, humans have drastically altered the diet that ruminants consume. The use of grain feed increases the productivity of animals and the economic benefit of animal husbandry. In our study, the diversity and richness was significantly lower in high dietary C:F ratios treatments than low dietary C: F ratios (HS1, HS2 and HS3). *Lin et al. (2015)* found that the Shannon diversity index of bacteria were higher in low concentrate diet group than in high concentrate diet group concentrate diet group (*Lin et al., 2015*). *Liu et al. (2015)* discovered the diversity of ruminal epithelial bacterial community from goats fed the hay diet were significantly higher than those fed the high-grain diet (*Liu et al., 2015*). Diet composition may effect on the diversity index of microorganisms.The increase in microbial diversity could be because low dietary concentrate diets provide a greater range of carbohydrate substrates (e.g., cellulose and many heteropolysaccharides) than high dietary concentrate treatments and/ or because microorganisms grow faster in high pH conditions (*Hobson & Stewart, 2012*).The growth performance results indicated that increases in the concentrate feed level from 0 to 45%, exerted a positive effect on the feed intake, BW gain, gain rate and $F_{CR}$ in Tibetan sheep. Therefore, based on our findings, we can conclude that improved growth performance is not linked to a higher diversity of rumen microorganisms. It is well known that a decrease in the bacterial diversity can induce the evolution of a certain bacterial group and thereby its dominance in the community. Thus, in our study, a decrease in bacteria diversity is associated with a higher dominance index (phyla *Bacteroidetes* and genera *Prevotella_1*), which ensured that the host could receive more nutrition and ultimately exhibits an improved growth performance. In addition, in our study, increases in the dietary C:F ratio from 0:100 to 30:70 were associated with increases in both the rumen bacterial diversity and the host growth performance, whereas, dietary C:F ratios of 45:55 and 60:40 could only promote growth performance of Tibetan sheep. *Briesacher et al. (1992)* observed that the digestive of the rumen is influenced by the number of bacterial species and the total abundance of bacteria. *Wanapat et al. (2003)* investigated the microorganisms in cattle and swamp buffaloes and observed that swamp buffaloes exhibit a stronger ability to digest cellulose than cattle due to their greater abundance of bacteria ($1.6 \times 10^9$/mL) compared with that of cattle ($1.36 \times 10^9$/mL). In our study, the HS3 group (C:F ratio 30:70) exhibited a richer microbial community and an increased in microbial diversity.Therefore, the inclusion of 30% concentrate in feed might be a good choice of diet that can be provided to Tibetan sheep. However, we did not perform digestibility experiments with all the treatments, which is a limitation of this study. Therefore, whether a rumen environment with a high bacterial richness and a high bacterial diversity is associated with a higher digestive capacity of Tibetan sheep remains to be investigated.

## CONCLUSION

The different dietary C: F ratios affected the growth performance, rumen fermentation and rumen bacterial diversity of Tibetan sheep. Increasing the dietary concentrate feed

level from 0% to 60% exerted a positive effect on the DMI, BW gain, gain rate and $F_{CR}$ of Tibetan sheep during late non-growing and green-up period, and Tibetan sheep fed with 45% concentrate level for barn feeding can result in a better improvement in animal performance. Moreover, high C:F ratios significantly increased ammonium nitrogen, reduced the A:P molar ratio and changed the composition of the bacterial community.

## ACKNOWLEDGEMENTS

We would like to thank Yinfa Ji for his help in sampling. We also thank Hanzhong Ji, senior livestock engineer from Haibei Prefecture, for providing the experiment site.

### Funding

This work was supported by the National Key R&D Plan (NO. 2016YFC0501905, NO. 2016YFC0501805), the "Strategic Priority Research Program" of CAS, the "Key Technology Support Program" of Qinghai province (No. 2018-S-2, 2015-SF-A4-2, 2016-NK-148, 2017-SF-A6, 2017-NK-153) and the Qinghai Innovation Platform Construction Project (No. 2017-ZJ-Y20) and NSFC (No. 31402120). The funders had no role in study design, data collection and analysis, decision to publish, or preparation of the manuscript.

### Grant Disclosures

The following grant information was disclosed by the authors:
National Key R&D Plan: 2016YFC0501905, 2016YFC0501805.
"Strategic Priority Research Program" of CAS, "Key Technology Support Program" of Qinghai province: 2018-S-2, 2015-SF-A4-2, 2016-NK-148, 2017-SF-A6, 2017-NK-153.
Qinghai Innovation Platform Construction Project: 2017-ZJ-Y20.
NSFC: 31402120.

### Competing Interests

The authors declare there are no competing interests.

### Author Contributions

- Hongjin Liu and Tianwei Xu conceived and designed the experiments, performed the experiments, analyzed the data, contributed reagents/materials/analysis tools, prepared figures and/or tables, authored or reviewed drafts of the paper.
- Shixiao Xu conceived and designed the experiments, authored or reviewed drafts of the paper, approved the final draft.
- Li Ma approved the final draft, dr Li Ma helped to collect the samples and review the manuscript.
- Xueping Han approved the final draft, dr Li Ma helped to review the manuscript.
- Xungang Wang, Xiaoling Zhang and Xinquan Zhao approved the final draft.
- Linyong Hu approved the final draft, dr Linyong Hu helped to collect the samples and review the manuscript.

- Na Zhao approved the final draft, dr Na Zhao helped to collect the samples and review the manuscript.
- Yongwei Chen approved the final draft, mr Yongwei Chen helped to collect the samples.
- Li Pi contributed reagents/materials/analysis tools, approved the final draft.

## Animal Ethics

The following information was supplied relating to ethical approvals (i.e., approving body and any reference numbers):

All animal care procedures were consistent with the provision of the Institution of Animal Care and the Ethics Committee of the Northwest Institute of Plateau Biology, Chinese Academy of Sciences (NWIPB20160302).

## Data Availability

The sequencing data for the 16S rRNA genes are publicly available in the NCBI Short Read Archive: PRJNA477411.

## Supplemental Information

Supplemental information for this article can be found online at http://dx.doi.org/10.7717/peerj.7462#supplemental-information.

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
