# Peer review of "Effect of dietary concentrate to forage ratio on growth performance, rumen fermentation and bacterial diversity of Tibetan sheep under barn feeding on the Qinghai-Tibetan plateau"

_PeerJ, doi:10.7717/peerj.7462_

## Round 0.1 · original submission · Major Revisions

Please provide a detailed rebuttal letter with clear responses to each of the reviewers' comments along with your revised manuscript.

Reviewer 1 ·

Basic reporting

Overview this paper is clear, well wrote, all tables and figures were adequate and literature references are enough.

Experimental design

The experimental design is correct, but the statistical analysis more appropriated on different levels of concentrate in diet is regression.

Validity of the findings

On discussion about rumen pH, the time of collection of the ruminal liquid after the supply of the concentrate and or diet is fundamental to verify the change in pH. The decrease in pH is associated with a high starch on diet and acid lactic production. The discussion on protein content seems to be inadequate. Because the bacteria need 7% of protein on diet, then only in the first treatment (HS1) the protein content was less than 7%. Regard to volatile fatty acids the term most accepted today would be short chain fatty acids. In addition, for the term of comparison between treatments the correct would be to present the data in proportion from the total and not in concentration. Necessary correct ml to mL on text

Additional comments

Considering the complex microbioma ruminal, this work is very important and contributes with new information about the interactions of the diet and microbiota

Reviewer 2 ·

Basic reporting

This study describes the potential effect of the diets varying in the ratio of concentrate to forage on the rumen fermentation, bacterial diversity and growth performance in Tibetan sheep. The study is interesting, and provides valuable insights to the ruminant nutrition community.

The English language should be improved. There are grammar errors throughout the manuscript.

Experimental design

1. need more explanation and power analysis on why 20 animals, with 4 in each treatment groups is sufficient.

2. lacks information on how the DNA concentration was measured.

Validity of the findings

findings are reasonable

1. more information is needed on the number of reads mapped to each OTU.
2. the authors stated in the discussion that the ruminal pH values in this study did not show significant differences, though previous study by Kennelly (1999) did observe a significantly decreased pH values. Not sure why the protein level was proposed as a possible explanation. Please clarify.

Additional comments

Major comments:
1. Line 76-77, is 20 animals total sufficient for this study? Please provide a power analysis.
2. Lines 120-121, How the DNA concentration was determined, aside from using the 2% agarose gel? As the correct measure of DNA concentration is critical to downstream sequencing library preparation and analysis
3. Line 187, What is the number of raw reads mapped to each OUT?
4. The measurement of ruminal fluid pH, the authors stated in the discussion that the ruminal pH values in this study did not show significant differences, though previous study by Kennelly (1999) did observe a significantly decreased pH values. Not sure why the protein level was proposed as a possible explanation. Please clarify.
5. The English language should be improved. Below are some examples (though not exhaustive)
a. Line 152, please use the sign “”
b. Line 268, please consider to remove the word “exist”
c. Line 269, please consider to remove “because of our”
d. Line 270, please rephrase the sentence “leading the protein level in diets were lower”
e. Line 281, please rephrase the sentence “the expected increase the molar proportion…”
f. Line 286, please rephrase the sentence “no difference among each treatments…”
g. Line 316, please consider to change “have a habit of” into “has a habit of”

Minor:
1. Line 251, please elaborate on what does “positively jointly” mean?
2. Lines 320-321, “the abundances increased”, is this significant? What is the p-value?

·

Basic reporting

There were a lot of typos and wording mistakes along the whole manuscript found, for example:
line 158 ("Wiener" not "Weiner"), line 216, 370, Figure 3: NOT "Winner";
line 6: high-throughput [] was applied
line 19 Tibetan sheep
31 delete "a developed strategy" or rewrite it
34 certain instead of some ; rarely instead of seldom, use instead of with
35 this is not correct: concentrate feed would be even higher in certain nutrients as grassland, so rewrite the sentence in terms of: "could be disadvantageous in terms of rumen health, rumen acidosis etc."
35-37: if you DID already experiments, please give reference; if this is a grammar mistake, you need to write "in this study .. we performed.."
38 Start the sentence with "Rumen microbes [...]
39-40: reassess sentence: the microbes digest the plant materials and provide nutrient to the host, they probably perform their digestive metabolism without the host itself, the host only provides a adequate habitat for the microbes
41 not only CP, ADF and NDF: all nutrients that are ingested by the host! here you can be more general in writing protein, complex carbohydrates, starch, sugars, fat..
43 delete "crude", its just bacterial protein;
49 However, only few
76 change "non antibiotics" to "not under current antibiotic treatment"
79 are shown in Table 1
81 forage; from the same local herdsman
88 grind throug a 1mm sieve
92-93 electric pump, which was confirmed to show similar results for rumen microorganisms and fermentation parameters as samplig via a rumen cannula

line 158 indexes and species
line 160 weighted Unifrac
line 180 Body weight, grain ratio and ADG (--> Abrreviations need to spelled out at the beginning of the sentence
line 187 Rarefaction curves
throughout the manuscript: use for P either lowercase or uppercase letters P / p, don't switch
194 the highest relative abundance
213 [..] phyla. The relative [...]
217 -218 rarrange sentence: Both parameters were significantly higher in [..]
243 fermentation parameters pH, A:P and the CP of the diet
251 what do you mean by "positively jointly influenced" - rewrite this sentence
253 I would rather use "correlated" instead of "influenced" because we don't know for sure which parameter was the driving one.
259 treatment
266 and diets
283-285 Rearrange sentence: From a energy-utilization point of view, [...] means This can also explain that the high-concentrate diet
287 treatment (singular!)
296 In the current study
305 what do you mean by older bovines? old in terms of age, old in terms of DIM? please be more concise!
323 gourp (singular!)
330 according to your results, Prevotella_1 and CP were both increasing, so it ist not an "undulation", please rechekc and better use "increase"
332-336 Pseudobutyrivibrio genus / spp., Prevotellaceae! Correct mistaken genera names. Why do you cite Yange et al 2010 at this point in the sentence, they found Pseudobutyrivibrio in their study, but did not isolate and describe it; you should use citations to explain your findings. Rearrange this or delete the citation.
344 don't use "altered" in a manuscript, be concise with "increase" or "decrease"!
356-361 This abstract is very vague and general, please rewrite and be more concise in relation to your findings
365-366 [...] was significantly higher in groups HS3, HS4, and HS5 than in HS2 and HS2 (P<0.05).
375-376 Rearrange the sentence [Therefore, ...] to for example: From our findings we can conclude that a better growth performance is not linked to a higher diversity of rumen microorganisms.
377-379: you cannot state that the microorganisms kept growing, you did not do any cultivation or counting; also correct for: when the C:F ratio ranged from 0:100 to 30:70, [..] With a C:F ratio of 45:55 and 60:40 the growth performance could be promoted.- suggesting that ...
lines 373, 388,395 - no figure and table references needed in the discussion section
387 there seems the beginning of the sentence missing: Therefore the inclusion of x% concentrate in the feed might be a good choice for [..], delete "than any of the other groups"
390 why do you only discuss cellulose digestive capacity? when the bacterial community is changing, all digestive processes are affected in the rumen.
390-391 delete first "growth performance", adapt sentence
395-398 rearrange sentence, as for example: Therefore, in our study the increasing abundance of B. and Pr. in the rumen of Tibetan sheep could possibly contribute to a higher nutrient availability for the host and consequently higher growht performance.
400 gain rate
401 you mention the cold season, please give more informations about the nutrient availablitiy during the warm season, would your findings also apply for the warm season?
403-404 that there is no better growth
407-408: Please name specific persons, institutes, teams and what they contributed!
236-237 [..] ruminal samples, being significantly lower in HS1 compared to the other HS groups.


A proper proof read by all authors is requested in order to correct for such kinds of writing mistakes! Moreover I suggest the proofreading of the manuscript by a native English speaker.

Please spell out all non-SI abbreviations at first usage in the main manuscript (line 5 HS, etc.); use the same nomeclature for the same meanings: as for "high concentrate" always use "high concentrate" etc.; this improves clarity about what you are talking about;

The introuction about the knowledge of rumen microbes and the influence of increasing amounts of concentrate feed is too general for this manuscript. It is well known that a change in diet changes the rumen microbes. 55-60: It is well known that NGS is used in studying rumen microbes, you don't need to state that. be more precice: as for example in line 44-46: do not write "useful products", which products? Propionate?. Focus on your special questions on Tibetan sheep. The gap of knowledge of rumen microbes in this sheep is only little supported by the background information given. The reviewer would like to see more informations about the specific region the sheep live in, their exact purpose, their normal growth performance, the health or management problems that occur with the grassland feeding, what you mean by scientific breeding (line 62), their special physiology, the cost effectiveness of a concentrate diet for the farmers to underline the need for reasearch in this field. Also differences of this sheep to other ruminants such as cows could underline the gap of knowledge. More references concerning studies with changes in diet in sheep are required (as for ex. Lettat et al, 2010 J. Anim. Sci.; Minuti et al 2014, Journal of Animal Science; M.M. Zein-Eldin et al, 2014, Biotechnology in Animal Husbandry; Sonia Andrés et al 2018,Small Ruminant Research, etc..)
The Tibetan sheep have been used in some studies about methane emmission, it would be worth looking in those studies for rumen fermentation parameters to discuss: Wang M et al., Front Microbiol 2016;7:850; Zhang et al. (2016) Curr. Biol. 26:1873–1879;

A clear Hypothesis is missing. Please state it!

Figures are tables: Altough the requirement for legends for figures and tables is optional in the Journal, at least the the feeding groups (HS) should shortly be described also in the figures so that they can be understand as standalone;
Table 4: The calculation of Spearman correlation needs also be mentioned in the statistical part in M&M of the manuscript. Rearragen the table, so that there are no blanc boxes;


Supplementary data: Table S2: change "nutrients" to "concentrate", Table S3: change "functional" to "selected" ; change "item" to "species",

Experimental design

There are many unclear analyses and statements along the manuscripts:

76-79 Did you perform a statistical power analysis to prove n=4 to be sufficient for this kind of research?

80 Table 1: give the exact ingredients of the concentrate feed; Please analyze the starch content of the fed diets. These parameters are important, since the most active nutrient according to concentrate feed is varying starch levels due to varying concentrate feedstuff. In M&M line 81 only "oat hay" is given as forage feed, in Table 1 also "Green hay" is stated. Please be clear along the manuscript what was exactly fed! Please define "green hay"
Also a separate analysis of the concentrate and the forage feed as supplemental material would improve the quality on a nutritional state of the manuscript, because oat hay also contains noteworth amounts of non-structural carbohydrates;

82 were the animals also housed separately? or only fed separately?
How much feed was offered? how was the feed mixed / processed? Who performed the feeding and the sampling?
74 and 84 which diet was fed during the 15 days adaptation? The same baselinediet for all groups, or the HS1-HS5 diets for each specific group?
90 for future studies I recommend to do weighting more often if you want to describe Average daily gain. because with n=4 you might have high variations, also state timepoint of weighting (before, after feeding?)
91 when exactly were the samples taken, before feeding, after feeding, which timepoint?
144-146 give more details about your quality filtering: which treshold did you use?
158 state all alpha diversity parameters you analysed. Also analyse for Goods_coverage, in order to prove sampling and sequence depth, which should be above 90% for each sample;
164 For stronger explanatory power of correlation I would recommend to do spearman correlation also for the microbiota and fermentation parameters instead of RDA plots.
165 why did you only perform analysis of the top 32 known genera?
167 State your spearman correlation analysis in the statistic section

182. what is the difference between DMI and DMI consume?
188 please give the minimum sequence number you rarefied your rarefaction curves together with the %
192 41 phyla (and also 129 genera as in line 200) seems like a very high number, did you remove singletons from your sequence downstream analysis? Also give your phyla, genus and species talbes as supplementary material
195 change "less" to "moderate"
194 Please give information how you calculated realtive abundance: if the highest 10 phyla made 98% of the sequences, how can then for example Lentisphaere still have an abundance of 5.4%??
201: ist it really 0.3%? should that not be more higheer if they are the most important bacteria?
203 This demonstrates variabilit [..] -- belongs to discussin section
Table 3, Table 4, M&M section: In the material and methods section (line 163) the calculation of the feed efficiency (calculated by output/input) is claimed, however the results given in Table 3 as feed (conversion) efficiency (FCE) seem to be results of feed conversion ratio (FCR) calculation (calculated as output/input)!. Please clarify in the M&M section which of these two parameters was calculated and how, and correct your results data according to that. Also review the values used for calculation of the spearman correlation between FCE/ FCR(?) and the other parameters, since a decreasing FCR should result in a negative spearman correalation value when analysed with the increasing DMI, increasing BW, and ADG (and not positive, as given in table 4) and vice versa with Shannon, Observed species and Chao1!
214 do you have a certain reference for Prevotella_1 or was this just a blast result? Beucase in the discussion (329 and others) you discuss this Prevotella_1 very generously. I do not agree that there is so much knowledge about what this Blast result can degrade or not in the rumen.
232: state in which diets Prevotella_1 exactly increases!
232 and others: give the reason why you have chosen for example the top 12 genera? what were your criterions?
339 fibrolytic

Validity of the findings

219: Be careful when interpreting your PcoA plot: the statement that all five groups were clearly distinguished is not supported by your plot: grouop HS1 and 2 cluster very closely together, so do HS4 and 5! and HS3 is ways more close to HS1 and 2 because PC2 represents only 7.14% of the variabilty. Reinterpret this section! also the numbers given in the text (line 222) do not fit to your figure!
316-318 Wang et al. 2016 did not describe any Acidobacteria, nor that they are inhibited in growth by proteins. Referring to general knowledge, dietary protein does not inhibit rumen bacteria.
366-368 You state that S.ruminatium was higher in the high-concentrate groups because they can withstand low ruminal pH and acid accumulation, but in your findings the pH and total VFA were not changing with concentrate feed. Please reconsider this discussion point!
393-395 You give the point that "although richness and diverstiy decrease" in your study, "Bacteroidetes increased" - this is not "although", it is well known that when diverstiy is decreasing, that certain bacterial groups can evolve, and so start to dominate the community, a decrease in diversity mostly goes hand in hand with a higher dominance index, and higher abundance of a certain group of bacteria.
270 Why is the protein level playing an important role in pH changes in the rumen? Starch would be the main driver for pH changes in the rumen. Please review this part and do proper literature research!
274-277 why is the rumen of Tibetan sheep special in terms of maintaining a stable pH? Please give better explanations for your findings! To my knowledge the general forestomach system is similar in all sheep.
287 "acute" is not appropiate for your research quetion: acute changes, including acute pH changes would clearly be seen in your samples. There is no acute change expected with a concentrate: forage ratio of maximum 60:40
290-292 a difference of 390 OTUs is not a noteworth difference between sequencing studies. remove or adapt this discussion point. also the reasons are very general. Do focus on the groups of bacteria found!
319 and following: Again: do not overinterpret protein related changes for prevotella. Better use correlation analysis for this data; prevotella can degrade everything than lignin and cellulose, they are not solely dependent on protein.
338- 343 the discussion about the ruminoccus species is not related to your research question. That R.albus and R.flavefaciens do compete in vitro is well known, but you need to discuss your findings in terms of an in vivo rumen study and to sheep production level
Figure 1: there seems to be a clustering of certain groups of sheep and the feeding groups, please state this in your results;

Additional comments

The authors investigated the effect of increasing replacement of the traditionally fed grass diet with concentrate based feed in Tibetan sheep. Analyses in respect to performance of the animals (DMI, BWG), rumen fermentation parameters (VFA, NH3-N) and rumen microbiome (16S rRNA gene sequencing) were performed. The trial setup is in the reviewers opinion sufficient to answer first questions of this research approach, analyses performed are well established methods in that field and the manuscript is well structured. However, there is a massive weakness in how the manuscript was written and results were interpreted. All the background information, description of the M&M and discussion are written too general, not focused on the research question, do have many unclear passages, writing typos and unclear, incoherent and weak descriptions of methods, results and discussion. I insist on revising the whole manuscript especially according to what is described above.

---

## Round 0.2 · Minor Revisions

Thank you for the adequate revision. Please proceed to the last minor corrections suggested by the reviewer.

Reviewer 2 ·

Basic reporting

The revision is adequate. However, the English language still needs improvement. Numerous grammar errors existed in the revised manuscript.

Experimental design

No comment.

Validity of the findings

no comment.

Additional comments

Comments
The revision is acceptable.

The English language still need improvement. Below are some examples (though not the exhaustive list):
In Abstract:
1. Line 30, “The results showed that increases dietary”, please consider to change “increases” to “increasing”
2. Line 32, “The increases dietary concentrate feed level”, please consider to change “increases” to “elevated” (or something similar)
3. Line 37, “In conclusion, increases dietary concentrate”, please consider to change “increases” to “increased”

In introduction:
1. Line 60, “Seriously, irrational grazing during this period”. Please consider to remove “Seriously”
2. Line 72, “can be affected by many factors such as diet”, please consider to revise the sentence into “can be affected by many factors, such as diet”

In Materials and Methods:
1. Line 186, please use appropriate sign for “≧97%”.

In Results:
1. Line 226, “The top ten phyla, which exhibited the highest relative abundances”, please consider to revised it to “The top ten phyla, which exhibited the highest relative abundances, were prevalent in all the samples, accounting for nearly 98% of all sequences.”
2. Line 231-232, please rephrase this sentence, “10 genera with relative abundances that represented more than 1% of bacterial community were perceived”

In Discussion:
1. Line 315, “This result might due to the”, please consider to change it to “This result might be due to the”
2. Line 319, “NH3-N concentration, which consisted with former study”, please consider to change it to “NH3-N concentration, which is consistent with former study”

---

## Round 0.3 · accepted · Accept

Thank you for the revision. Your paper is now accepted for publication in PeerJ.